

# Microbial community in buckwheat rhizosphere with different nitrogen application rates

Dongsheng Wang and Haike Ren

College of Life Science, Shanxi Normal University, Taiyuan, Shanxi, China

## ABSTRACT

Microorganism plays a pivotal role in regulating sustainable development of agriculture. The excessive application of nitrogen fertilizer is considered to affect the microbial structure in many agricultural systems. The present study aimed to assess the impacts of nitrogen application rate on microbial diversity, community and functionality in rhizosphere of Tartary buckwheat in short-time. The nitrogen fertilizer was applied at rates of 90 kg (N90), 120 kg (N120) and 150 kg (N150) urea per hectare, respectively. The soil properties were measured chemical analysis and displayed no difference among treatments. Metagenome analysis results showed that the microbial diversity was not affected, but the microbial community and functionality were affected by the nitrogen application rate. According to the Linear discriminant analysis effect size (LEfSe) analysis, 15 taxa were significantly enriched in the N120 and N150 groups, no taxon was enriched in the N90 group. Kyoto Encyclopaedia of Genes and Genomes (KEGG) annotation results revealed that the genes related to butanoate and beta alanine metabolism were significantly enriched in the N90 group, the genes related to thiamine metabolism, lipopolysaccharide biosynthesis and biofilm formation were significantly enriched in the N120 group, and the genes related to neurodegenerative disease was significantly enriched in the N150 group. In conclusion, short-time nitrogen fertilizer application shifted the microbial community structure and functionality.

## INTRODUCTION

Tartary buckwheat (*Fagopyrum tataricum*) is a dicotyledonous plant belong to the family Polygonaceae. It is originated from southwest China and now is planting in many countries, for the nutritional and health-benefitting quality. Its gluten-free grains have rich essential nutrients, as well as bioactive compounds such as flavonoids and polyphenol (*Zhu, 2016*; *Zhong et al., 2016*). Nitrogen is one of the most important nutrients that affect crop growth and productivity (*Ren et al., 2022*). Although it is a barren-tolerant crop, the application of suitable nitrogen fertilizer was proved to improve the yield and quality of Tartary buckwheat in last several decades (*Zhang et al., 2019*). Nitrogen fertilizer can regulate the synthesis and accumulation of starch in grains (*Gao et al., 2021*). The starchy endosperm provides nitrogen sources for cotyledon development in the process of buckwheat seed development (*Liu et al., 2018*). However, excessively applied mineral nitrogen fertilizer is a waste of

Corresponding author
Dongsheng Wang,
wangds@sxnu.edu.cn

nitrogen resources and may cause soil acidification, compaction, and contamination, which is harmful to the soil ecology (*Zhang et al., 2021*). Therefore, it is urgent to improve the use efficiency of nitrogen fertilizer and diminish its side-effects.

The rhizosphere is the zone of soil that closely associates with living plant roots. It is governed by complex interactions between plants and microorganisms. These interactions have great influence on plant growth and productivity. Many rhizosphere microorganisms can enhance the formation of stable soil aggregates, improve soil nutrition status, decompose soil organic matter (*Luan et al., 2020*). Others can prevent plants from the damage of pathogens. On the other hand, plants can alter the microbial structure and diversity in rhizosphere, by root activities and exudates.

Human agricultural activities, including mineral fertilizers application, also affect the soil microbial diversity and functionality associated with the crop rhizosphere (*Essel et al., 2019*; *Nakayama & Tateno, 2022*). For instance, nitrogen fertilizer could regulate the microbial community *via* influencing the plant community and acidifying soil (*Zeng et al., 2016*). Increasing soil nitrogen availability could inhibit the microbial enzyme activities and alter the community structure (*Paungfoo-Lonhienne et al., 2015*). In early rice paddy, mineral nitrogen fertilizer reduced the abundances of methanogenic and methanotrophic bacteria (*Liu et al., 2019*). Application of nitrogen fertilizer increased the relative abundance of genes related to DNA/RNA replication, electron transport, protein and carbohydrate metabolism (*Fierer et al., 2012*; *Leff et al., 2015*). However, the impact of nitrogen application on microbial diversity and functionality in buckwheat rhizosphere is still unclear.

Soil metagenomic sequencing is a widely used powerful tool to evaluate microbial diversity, as well as functional attributes (*Orellana et al., 2018*). In this study, we explored microbial community, diversity and functionality in Tartary buckwheat rhizosphere with different nitrogen fertilizer application rates, based on metagenomic analysis. The aim was to assess the side-effect of excessive application of nitrogen fertilizer on soil ecology in Tartary buckwheat field.

## MATERIALS & METHODS

### Soil sampling

The one-year field experiment was carried out using a randomized complete block design at Mengjiazhuang (37°43′N, 112°59′E), Jinzhong City, Shanxi Province, China. Buckwheat cultivar Jinqiao no. 5 was planted with a density of 90 seeds per meter square. The growth period was from the late May to the beginning of October. Fertilizers were applied at seeding time. Both of phosphate and potassium fertilizers were applied at a rate of 90 kg per hectare, as the form of $Ca(H_2PO_4)_2$ and KCl, respectively. Nitrogen fertilizer was applied at three rates: 90 (N90), 120 (N120) and 150 (N150) kg granular urea per hectare. Five-points sampling method was performed to collect soil samples when harvesting.

### Measurement of soil chemical properties

Soil samples were air-dried, grinded and sieved. The pH was tested by a pH meter (Leici, Shanghai, China). The total nitrogen content was measured by Kjeldahl method using a

Kjeldahl nitrogen analyzer (Elementar, Langenselbold, Germany). The available nitrogen was detected by alkaline hydrolysis diffusion method (*Finzi et al., 2015*).

## DNA extraction and metagenomic sequencing

The total DNA in soil samples were extracted using a DNeasy PowerSoil Kit (QIAGEN, Hilden, Germany), according to the manufacturer's instructions. Agarose gel electrophoresis and a NanoDrop ND-1000 spectrophotometer (Thermo Fisher Scientific, Waltham, MA, USA) were used to evaluate the quality and quantity of the extracted DNA, respectively. Quality-controlled DNA was shared to construct 400 bp (insert size) paired-ended libraries using a TruSeq DNA Nano High Throughput Library Preparation Kit (Illumina, San Diego, CA, USA), following the manufacturer's instructions. The libraries were sequenced on an Illumina HiSeq X-ten platform (Illumina, San Diego, CA, USA) by Personal Biotechnology Co., Ltd. (Shanghai, China). All sequences are available in NCBI under accession no. PRJNA939946.

## Sequencing data analysis

Raw sequencing data was saved in FASTQ format and firstly filtered to discard the reads with length <50 bp or contain ambiguous bases. Specifically, the adapters were trimmed off using Cutadapt (v1.2.1), and the low-quality reads (<Q20, read accuracy <99%) were removed by a 5-bp sliding window in fastp (v0.20.0). The high-quality clean reads were de novo assembled to construct the metagenome by MEGAHIT (v1.0.5). All coding sequences (CDS) longer than 300 bp were predicted using MetaGeneMark (v3.25) and clustered by CD-HIT (v4.8.1) at 95% amino acid sequence identity and 90% coverage. Gene abundance was estimated by SOAPdenovo2 (v1.0) based on the counts of aligned sequences. Based on the LCA (lowest common ancestor) algorithm, taxonomy of the non-redundant sequences was annotated in NCBI-NT database by BLASTN ($e < 0.00001$) in Blast2lca software. The functional genes were annotated in KEGG, GOSlim, carbohydrate-active enzymes (CAZymes) and evolutionary genealogy of genes: Nonsupervised Orthologous Groups (eggNOG) databases using the DIAMOND alignment algorithm. Linear discriminant analysis effect size (LEfSe) analyses were used for screening the markedly enriched taxa and functional genes in each treatment by Galaxy (http://huttenhower.sph.harvard.edu/galaxy/). The Quantitative Insights Into Microbial Ecology (QIIME, v1.8.0) pipeline and R software (version 3.6.1; *R Core Team, 2019*) were used to analyze and visualize the sequencing data.

## RESULTS

### Chemical properties and microbial diversity in soil samples

The pH, available nitrogen and total nitrogen contents were ranged from 7.85 to 7.88, 1.98 to 2.03 mg/kg, and 1.35 to 1.38 g/kg, respectively, in tested soil samples. All of the three indices showed no significant difference (*t* test, $df = 2$, $P > 0.05$) among treatments (Table S1), indicating that the application rates of urea from 90 to 150 kg per hectare have similar influence on the soil chemical properties in rhizosphere of Tartary buckwheat.

A range from $8.3 \times 10^9$ to $10.0 \times 10^9$ filtered high-quality reads were obtained from the nine samples by an Illumina NovaSeq sequencing platform (Table S2). A total of 33,400
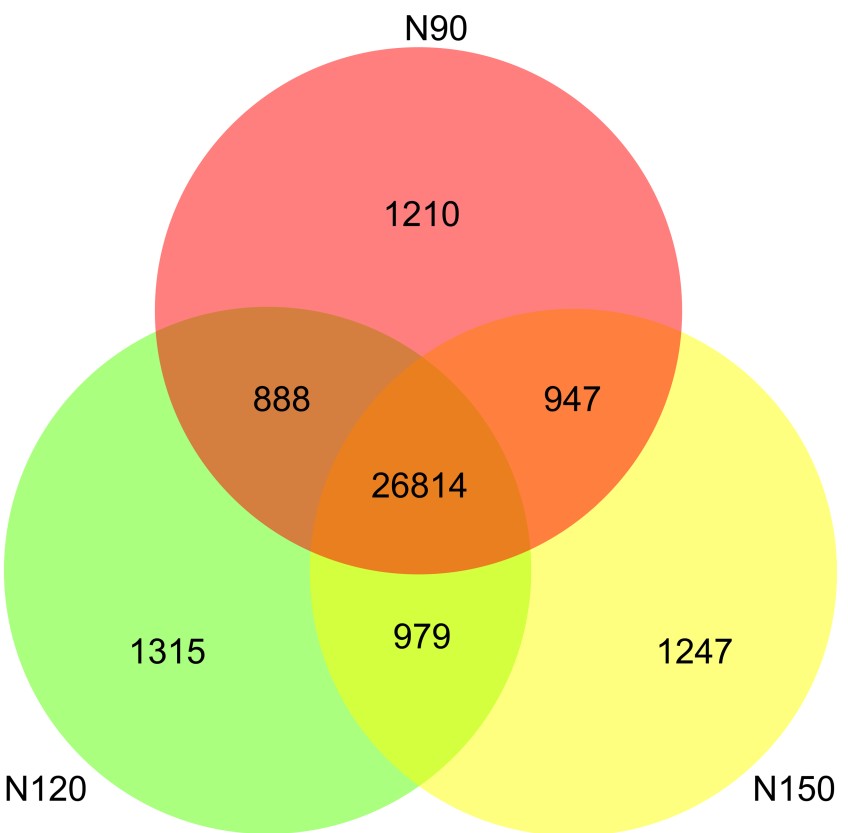

**Figure 1  Species numbers in rhizosphere of buckwheat with different nitrogen application rates.**

species were detected in all samples. Specifically, about 80% of the species were commonly appeared in the three groups. There were 1,210, 1,315 and 1,247 species only appeared in N90, N120 and N150, respectively (Fig. 1), implying that these species were sensitive to urea.

Nitrogen application rates showed no significant influence on the alpha diversity, including Simpson, Chao1, ACE and Shannon indices, of soil microorganisms ($t$ test, $df = 2$, $P > 0.05$) (Table S3). Furthermore, beta diversity was evaluated by principal components analysis (PCA), principal coordinates analysis (PCoA), nonmetric multidimensional scaling (NMDS) and unweighted pair-group method with arithmetic means (UPGMA). The results revealed that the three groups cannot separate individually (Fig. S1). Therefore, we concluded that the microbial diversity was not affected by the application rates of urea in rhizosphere of Tartary buckwheat.

## Microbial community structure and composition

Prokaryotic species was much more abundant than eukaryotic species in rhizosphere of Tartary buckwheat. As shown in Fig. 2, about 77% of the high-quality reads were blasted to bacteria and 22% of the reads were blasted to unclassified non-viral species. Only 0.3% of the reads were blasted to eukaryote. Across treatments, the relative abundance of top 20

Peer J

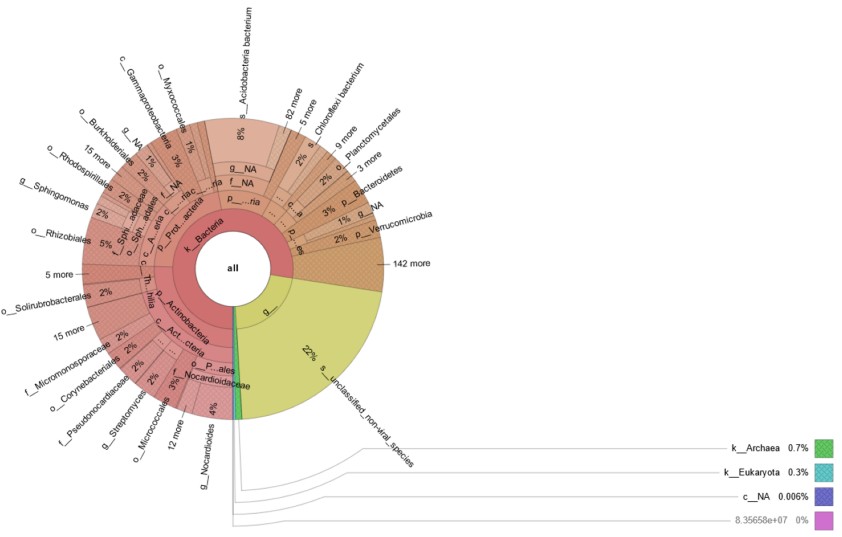

**Figure 2** **Microbial community composition in buckwheat rhizosphere.**

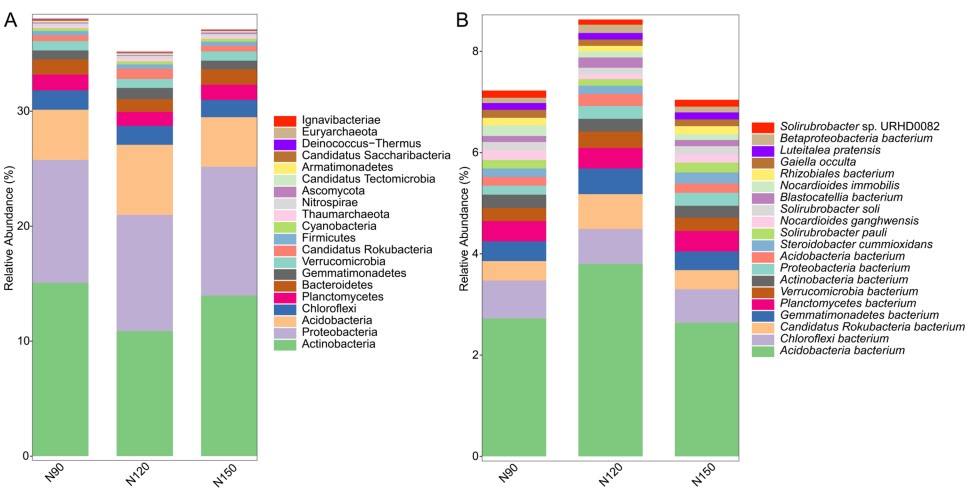

**Figure 3** **Top 20 taxa in buckwheat rhizosphere at the phylum (A) and species (B) levels.**

phylum and species were 35.2%–38.1% and 23.2%–29.3%, relatively (Fig. 3). The dominant phylum with relative abundance >1% were Actinobacteria, Proteobacteria, Acidobacteria, Chloroflexi, Planctomycetes, Bacteroidetes. Among the three treatments, the relative abundance of phylum Actinobacteria and Candidatus Saccharibacteria were highest in the N90 group. The relative abundance of phylum Acidobacteria, Gemmatimonadetes and Candidatus Rokubacteria were highest, while the relative abundance of Bacteroidetes were lowest in the N120 group. The relative abundance of phylum Ascomycota was highest in the N150 group.

At the species level, the dominant species with relative abundance >1% were *Acidobacteria bacterium*, *Chloroflexi bacterium*, *Candidatus Rokubacteria bacterium*,

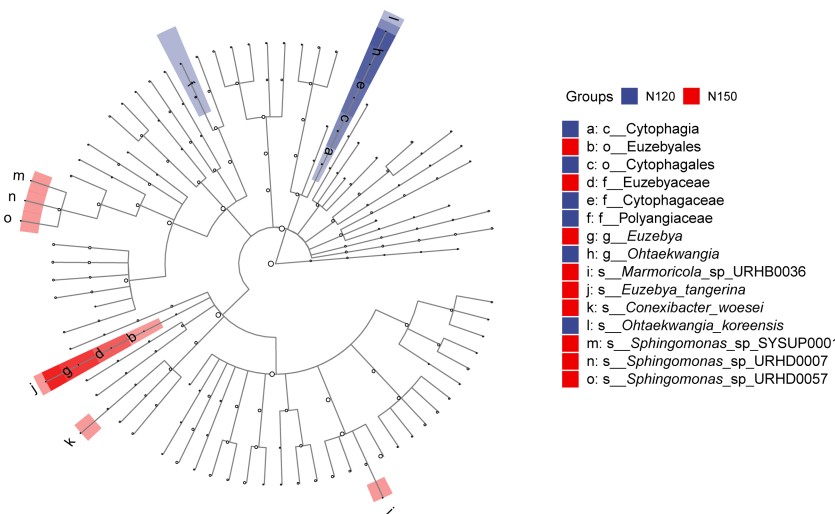

**Figure 4 Comparison of taxa in buckwheat rhizosphere with different nitrogen application rates based on LEfSe analyze.** The node size relates to the relative abundance of the taxon.

*Gemmatimonadetes bacterium* and *Planctomycetes bacterium*. In the N90 treatment, the relative abundance of *Proteobacteria bacterium*, *Nocardioides immobilis* were higher than those of other treatments. In the N120 treatment, the relative abundance of *Acidobacteria bacterium*, *Candidatus Rokubacteria bacterium*, *Gemmatimonadetes bacterium*, *Verrucomicrobia bacterium* and *Blastocatellia bacterium* were higher, while the relative abundance of *Nocardioides ganghwensis*, *Solirubrobacter soli*, *Solirubrobacter* sp. URHD0082 were lower than those of N90 and N150 treatments. In the N150 treatment, *Steroidobacter cummioxidans* and *Solirubrobacter pauli* were more abundant than those in the other two treatments.

LEfSe was applied to analyze the feature taxa markedly affected by nitrogen application rates (Fig. 4). Based on the LDA method (log 10 >2) and Kruskal-Wallis (KW) test, 15 taxa were screened in the N120 and N150 groups, while no taxon was screened in the N90 group. Specifically, class Cytophagia was significantly enriched ($df = 2$, $P = 0.03$) in the N120 group. Order Cytophagales was significantly enriched ($df = 2$, $P = 0.03$) in the N120 group, while Euzebyales was significantly enriched in the N150 group. Families Cytophagaceae and Polyangiaceae were significantly enriched ($df = 2$, $P = 0.04$) in the N120 group, while Euzebyaceae was significantly enriched ($df = 2$, $P = 0.04$) in the N150 group. Genera *Ohtaekwangia* was significantly enriched ($df = 2$, $P = 0.04$) in the N120 group, while *Euzebya* was significantly enriched ($df = 2$, $P = 0.04$) in the N150 group. At the species level, *Ohtaekwangia_koreensis* was significantly enriched ($df = 2$, $P = 0.04$) in the N120 group, while *Marmoricola* sp. URHB0036, *Euzebya tangerine*, *Conexibacter woesei*, *Sphingomonas* sp. SYSUP0001, *Sphingomonas* sp. URHD0007 and *Sphingomonas* sp. URHD0057 were significantly enriched ($df = 2$, $P = 0.03$ or 0.04) in the N150 group. It is reasonable to assume that high nitrogen application rate benefits the above-mentioned microorganisms.

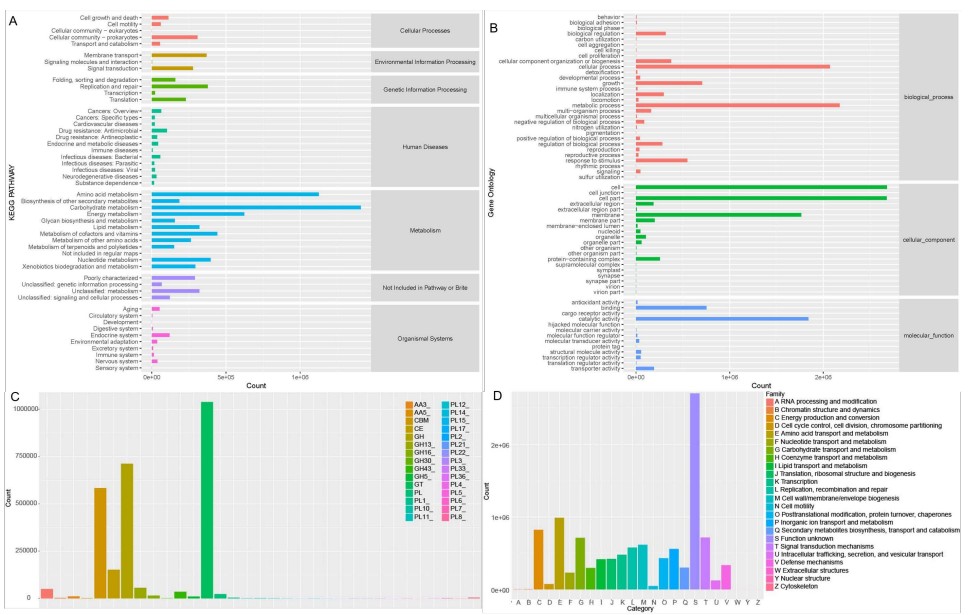

**Figure 5** Microbial functionality analysis in four databases, KEGG (A), GO (B), CAZyme (C) and eggnog (D).

## Microbial functionality and metabolic pathways

Soil microorganisms regulate soil properties and affect plant growth by various functions. In order to fully understand the microbial functionality in rhizosphere of Tartary buckwheat, we annotated the non-redundant proteins in KEGG, GOSlim, CAZymes and eggNOG databases (Fig. 5).

Generally, KEGG annotation results showed that microbial metabolic pathways were mostly enriched in metabolism at the first level, and mostly enriched in amino acid, carbohydrate, lipid, cofactors and vitamins metabolism and xenobiotics biodegradation at the second level in buckwheat rhizosphere (Fig. 5A). GO analysis revealed that the genes with high relative abundance were related to cellular process, metabolic process and growth at the biological process level, related to cell, cell part and membrane at the cellular component level, and related to catalytic activity, binding and transporter activity at the molecular function level (Fig. 5B). The top five CAZyme families annotated were GT, GH, CBM, CE and GH13 (Fig. 5C). EggNOG annotation results showed that function of about 25% of the proteins was unknown and the top five annotated functions were amino acid transport and metabolism, energy production and conversion, carbohydrate transport and metabolism, signal transduction mechanism, and replication, recombinant and repair (Fig. 5D). Combining the annotation results from the four databases, we found that metabolism and transport of various nutrients is a pivotal function for soil microorganisms in rhizosphere of Tartary buckwheat.

## Difference of microbial functionality among treatments

To clarify the impact of nitrogen application rates on microbial functionality, the difference among the three tested treatments was compared. A total of 320, 599 and 85,429 functions were annotated in KEGG, CAZymes and eggNOG databases, respectively. More than 90% of the functions annotated in KEGG and CAZymes databases, and over 80% of the functions annotated in eggNOG database were commonly appeared in all groups. In the KEGG database, there were three, five, and five functions only appeared in N90, N120 and N150, respectively. In CAZymes database, there were three, seven, and five functions only appeared in N90, N120 and N150, respectively. In eggNOG database, there were 1,946, 2,209 and 3,377 functions only appeared in N90, N120 and N150, respectively (Fig. S2). Pairwise comparison results showed that there was no significant difference ($t$ test, $df = 2$, $P > 0.05$) among the three treatments in both KEGG and CAZymes databases. In the eggNOG database, the relative abundance of genes annotated to transcription was significantly lower ($t$ test, $df = 2$, $P = 0.0001$), while the relative abundance of genes annotated to inorganic ion transport ($df = 2$, $P = 0.001$) and metabolism ($df = 2$, $P = 0.003$) was significantly higher in the N120 group, compared to those in the N90 group. There was no significant difference between N90 (or N120) and N150 ($t$ test, $df = 2$, $P > 0.05$) (Table S4).

The feature annotated functions markedly affected by nitrogen application rates was analyzed by LEfSe, on the basis of LDA method (log 10 >2) and KW test (Fig. 6). The results showed that in the KEGG database, functions of butanoate and beta alanine metabolism were significantly enriched ($df = 2$, $P = 0.04$) in the N90 group. Functions of thiamine metabolism, lipopolysaccharide biosynthesis and biofilm formation were significantly enriched ($df = 2$, $P = 0.03$ or $0.04$) in the N120 group. Function of neurodegenerative disease was significantly enriched ($df = 2$, $P = 0.04$) in the N150 group. Based on these results, we suppose that rhizosphere microorganisms provide nitrogen nutrient for Tartary buckwheat by hydrolyze nitrogenous compounds when urea applied with low rates.

In the eggNOG database, functions of replication, recombination and repair, energy production and conversion, information storage and processing, transcription, ENOG4105CBC, ENOG4105C53, ENOG4105CG3, ENOG4107QM5, ENOG4105C8T, ENOG4105C07 and ENOG4105C9C were significantly enriched ($df = 2$, $P = 0.03$ or $0.04$) in the N90 group. Function of ENOG4105C7B was significantly enriched ($df = 2$, $P = 0.04$) in the N120 group. No function was significantly enriched in the N150 group. In the CAZymes database, families of GH13, GT51, GH101, GT13, GH55 and GH73 were significantly enriched ($df = 2$, $P = 0.03$ or $0.04$) in the N90 group. GH31 family was significantly enriched ($df = 2$, $P = 0.04$) in the N120 group. No family was significantly enriched in the N150 group.

## DISCUSSION

Soil physicochemical characteristics are important indexes to evaluate soil quality. Over the last several decades, increasing chemical fertilizers especially nitrogen fertilizers were applied for higher productivity in China (Zhuang et al., 2022). The effect of nitrogen

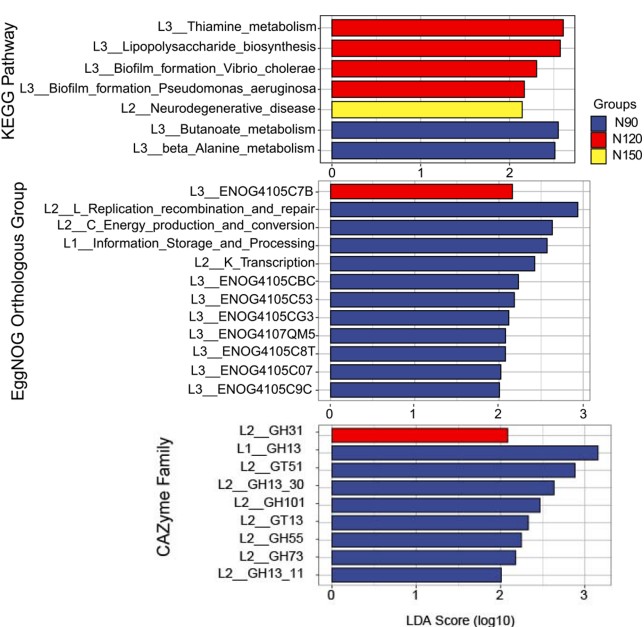

**Figure 6** LEfSe analysis of microbial functionality among the three nitrogen application rate treatments.

fertilizer on soil physical, chemical and biochemical properties are varied with fertilizer type and application rate (*Paungfoo-Lonhienne et al., 2015*). Previous studies proved that pH was apparently affected by nitrogen fertilizer application, partially due to the release of H ions during hydrolysis (*Schroder et al., 2011*). Among the commonly used nitrogen fertilizer sources, $(NH_4)_2SO_4$ showed highest impact on soil acidity, while urea showed lowest impact (*Sainju et al., 2015*). To take the trial period into account, we defined the trials last for more than five years as "long-term trial" and those less than five years as "short-term trial". In this study, the soil pH, as well as contents of total nitrogen and available nitrogen, were not affected by the nitrogen application rate. The possible reasons maybe the experiment was conducted in short-time and the nitrogen source was urea.

Generally, fertilization management system can impact the soil microbial diversity (*Rodríguez-Berbel et al., 2020*), but the impacts are frequently complicated and lack consistency. The effects of nitrogen fertilization on microbial diversity are likely site-dependent. For example, a meta-analysis showed that nitrogen fertilizer negatively affected the alpha diversity in agricultural soils (*Zhou, Wang & Luo, 2020*). However, fertilizers application increased the fungal abundances in the black soils of northeast China (*Hu et al., 2017*). *Fierer et al. (2012)* found that nitrogen application decreased the bacterial diversity in an agricultural system but showed no influence in grassland. The results in this study revealed that applying urea at rates from 90 to 150 kg per hectare does not affect the alpha and beta diversity of soil microbes in short-time.

Soil microbial community is supposed susceptible to the alteration of soil physicochemical properties, including pH, soil moisture, and nitrogen content (*Zhalnina et al., 2015*). Previous studies revealed that long-term application of excess nitrogen

fertilizer dramatically shifts microbial community structures (*Avio et al., 2013*; *Jach-Smith & Jackson, 2018*). Consistently, the short-time application of urea in this research changed the microbial community structures, and the effects were varied with the application rate. Across the three application rates, the relative abundance of Actinobacteria and Candidatus Saccharibacteria were highest in the N90 group, while the relative abundance of Acidobacteria, Gemmatimonadetes and Candidatus Rokubacteria were highest the N120 group, and the relative abundance of phylum Ascomycota was highest in the N150 group. The impacts of nitrogen addition on soil microbial community possibly are caused by its nutrient property, or by the alteration of soil environment (*Klironomos et al., 2011*). In this study, the soil properties were not influenced by the nitrogen application rate, indicating the shift of microbial community was caused by the nutritional attribute of nitrogen.

Nitrogen showed varied effect on microbial functionality with the fertilizer source and soil type. $NH_4NO_3$ application increased the abundance of DNA/RNA replication, electron transport, and protein metabolism-related genes (*Fierer et al., 2012*). Released-urea increased the relative abundances of genes associated with carbohydrate metabolism (*Leff et al., 2015*). Addition of urea showed no significantly different microbial activity in dry land cultivated soil (*Stark et al., 2007*). However, our research revealed that in buckwheat rhizosphere, application of urea at different rates have varied impacts on soil microbial functionality especially on genes related to butanoate, beta alanine and thiamine metabolism, lipopolysaccharide biosynthesis, biofilm formation, and neurodegenerative disease. Low nitrogen application (N90 and N120) increased the abundance of genes involved in nitrogenous compounds metabolism.

The effect of nitrogen fertilizer is also varied with crop species. In rhizosphere of sugarcane, application rates of urea modified the composition but not the taxon richness of fungal communities. The relative abundance of phylum Ascomycota was higher in high urea dose conditions (200 kg per hectare) compared to low urea dose (40 kg per hectare) (*Paungfoo-Lonhienne et al., 2015*). Consistently in this study, urea application rates had no influence on diversity but impacted the community structure of microorganisms, and the phylum Ascomycota was most abundant in N150 group. However, some studies of other species revealed that the influence of nitrogen fertilizer on microbial diversity is depend on the application rates. For example, nitrogen fertilizer rate significantly altered the bacterial diversity in maize rhizosphere (*Zhu, Vivanco & Manter, 2016*; *Zhang et al., 2020*).

## CONCLUSIONS

The nitrogen fertilization management shows varied impact on soil microbial diversity with the trial period, location, nitrogen source and crop species. Previous studies showed that long-term nitrogen fertilizers application altered soil properties especially pH, microbial diversity and community structure (*Schroder et al., 2011*; *Jach-Smith & Jackson, 2018*; *Zhou, Wang & Luo, 2020*). However, short-time application of urea showed no impact on the soil nutritious status and microbial diversity, but shifted the microbial community structure in rhizosphere of Tartary buckwheat in this study. As for the soil microbial functionality, the urea application rates affected the genes related to butanoate, beta alanine and thiamine

metabolism, lipopolysaccharide biosynthesis, biofilm formation, and neurodegenerative disease. Nitrogenous compounds metabolism was also activated by low urea application rates.

### Funding

This research was supported by the Fundamental Research Program of Shanxi Province (202103021224256) and the Scientific and Technological Innovation Programmes of Higher Education Institutions in Shanxi (2019L0462). The funders had no role in study design, data collection and analysis, decision to publish, or preparation of the manuscript.

### Grant Disclosures

The following grant information was disclosed by the authors:
Fundamental Research Program of Shanxi Province: 202103021224256.
the Scientific and Technological Innovation Programmes of Higher Education Institutions in Shanxi: 2019L0462.

### Competing Interests

The authors declare there are no competing interests.

### Author Contributions

- Dongsheng Wang conceived and designed the experiments, performed the experiments, analyzed the data, prepared figures and/or tables, authored or reviewed drafts of the article, and approved the final draft.
- Haike Ren performed the experiments, analyzed the data, authored or reviewed drafts of the article, and approved the final draft.

### DNA Deposition

The following information was supplied regarding the deposition of DNA sequences:
All sequences are available in NCBI under accession no. PRJNA939946.

### Data Availability

Raw data are available in the Supplemental Files.

### Supplemental Information

Supplemental information for this article can be found online at http://dx.doi.org/10.7717/peerj.15514#supplemental-information.

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
