# Peer review of "Microbial community in buckwheat rhizosphere with different nitrogen application rates"

_PeerJ, doi:10.7717/peerj.15514_

## Round 0.1 · original submission · Major Revisions

Dear Authors,

Thank you for submitting your work to PeerJ. Two reviewers have now gone through your manuscript and both of them suggested major revisions for the study. Please check their comments and suggests and revise your manuscript carefully. We look forward to seeing your revised version of the manuscript.

Kind regards,
Liang Wang, PhD
Academic Editor

·

Basic reporting

no comment

Experimental design

no comment

Validity of the findings

no comment

Additional comments

Microorganism plays a pivotal role in regulating sustainable development of agriculture. This manuscript studied the effects of nitrogen application on microbial diversity, community structure and function in the rhizosphere of Tartary buckwheat over a short period of time. However, there are still some issues that need to be resolved before the manuscript can be received.

1.Please provide the breed number of jinqiao5 to confirm its germplasm.

2.Tartary buckwheat is a small crop tolerant to barren. In the background, please fully explain the reasons for studying the effect of nitrogen application on Tartary buckwheat rhizosphere microorganisms in a short time and quote the relevant literature.

3,The description of the result part is too general to be analyzed according to the characteristics and experimental design of Tartary buckwheat species.

4.This manuscript has obtained many interesting results through metagenomics. However, in the discussion part, it needs to be compared with existing relevant studies to find species-specific results and universal rules between species, so as to highlight the significance of this study.

·

Basic reporting

At first, I would to mention the actual topic of the study – to compare the soil microbial community with impact of different nitrogen fertilizer application rate.
Bioinformatics approaches described in the manuscript can be improved. There are strong reasons to use amplicon sequence variants (ASVs) instead of OTUs for less noise and avoid spurious taxons (https://www.mdpi.com/2306-5354/9/4/146).
Callahan, McMurdie and Holmes points that “the improvements in reusability, reproducibility and comprehensiveness are sufficiently great that ASVs should replace OTUs as the standard unit of marker-gene analysis and reporting” Callahan, B., McMurdie, P. & Holmes, S. Exact sequence variants should replace operational taxonomic units in marker-gene data analysis. ISME J 11, 2639–2643 (2017). https://doi.org/10.1038/ismej.2017.119
L20: Metagenomic analysis cannot evaluate soil properties, the chemical analysis does
The number of species at Figure 1 presented with invalid scale, I suggest to use DeepVenn to build more precise diagram.

Experimental design

Some details of the study must be clarified to better understanding and reproducibility:
L74: Granular or prilled form of urea was used?
L93: how raw data quality filtering (FASTQ) was performed?
L105: What protocol and the reference database was used for taxonomic analysis in QIIME?
As said above, using OTUs approach produces a lot of spurious OTUs. L113 shows thousands of OTUs, most of which, I propose, share less than 1%. Such classification is hard to interpret. For further study see, f.e.:
https://doi.org/10.1371/journal.pone.0227434
https://doi.org/10.1038/s43705-021-00033-z

Validity of the findings

The Discussion section does not define the “long-term” and “short-term” periods of nitrogen fertilizer (L225, L228).
The changes of functional genes (KEGG) presented in Results not mentioned in Discussion too.
My notes about the main text:
• Some captions at Figure 5 are unreadable in manuscript
• Taxa with abundance less than 1% are non-informative at Figure 2 and might be omitted

---

## Round 0.2 · Minor Revisions

Please follow the reviewer's comments and revise the manuscript accordingly. When revising your manuscript, please highlight the changes in your manuscript so the reviewer and I could see the changes clearly.

·

Basic reporting

no comment

Experimental design

no comment

Validity of the findings

no comment

Additional comments

no comment

·

Basic reporting

pass

Experimental design

pass

Validity of the findings

Conclusion section is still very short and doesn't cover the long-term period of application. Many readers (so am I) go straight to the conclusion after reading the abstract.

Additional comments

Although authors response that Figure 1 was revised, I see no changes in the Venn diagram.

---

## Round 0.3 · accepted · Accept

I am pleased to inform you that after two rounds of peer review, your manuscript is now acceptable. Thank you for your contribution to the field.